

# Risks and opportunities for a Swiss hydropower company in a changing climate

Kirsti Hakala[1], Nans Addor[2], Thibault Gobbe[3], Johann Ruffieux[3], Jan Seibert[1,4]

[1]Department of Geography, University of Zurich, Zurich, 8057, Switzerland
[2]Climatic Research Unit, School of Environmental Sciences, University of East Anglia, Norwich, NR4 7TJ, United Kingdom
[3]Energy Board, Groupe E SA, Granges-Paccot, 1763, Switzerland
[4]Department of Aquatic Sciences and Assessment, Swedish University of Agricultural Sciences, Uppsala, 750 07, Sweden

*Correspondence to*: Kirsti Hakala (kirsti.hakala@geo.uzh.ch)

**Abstract.** Anticipating and adapting to climate change impacts on water resources requires a detailed understanding of future hydroclimatic changes and of stakeholders' vulnerability to these changes. However, climate change impact studies are often conducted at a spatial scale that is too coarse to capture the specificity of individual catchments, and more importantly, the changes they focus on are not necessarily the changes most critical to stakeholders. While recent studies have combined hydrological and electricity market modeling, they tend to aggregate all climate impacts by focusing solely
on reservoir profitability, and thereby provide limited insights into climate change adaptation. Here, we collaborated with Groupe E, a hydropower company operating several reservoirs in the Swiss pre-Alps and worked with them to produce hydroclimatic projections tailored to support their upcoming water concession negotiations. We started by identifying the vulnerabilities of their activities to climate change and then together chose streamflow and energy indices to characterize the associated risks. We provided Groupe E with figures showing the projected climate change impacts, which were refined over
several meetings. The selected indices enabled us to simultaneously assess a variety of impacts induced by changes on i) the seasonal water volume distribution, ii) low flows, iii) high flows, and iv) energy demand. We were hence able to identify key opportunities (e.g., the future increase of reservoir inflow in winter, when electricity prices are historically high) and risks (e.g., the expected increase of consecutive days of low flows in summer and fall, which is likely to make it more difficult to meet residual flow requirements). This study highlights that the hydrological opportunities and risks associated with
reservoir management in a changing climate depend on a range of factors beyond those covered by traditional impact studies. We also illustrate the importance of identifying stakeholder needs and using them to inform the production of climate impact projections. Our user-centered approach is transferable to other impact modeling studies, in the field of water resources and beyond.



## 1 Introduction

Hydropower is the most widely used renewable energy resource across the globe (Schaefli, 2015). Given this global importance, there is a growing need to support the adaptation of hydropower facilities and operations to changes induced by climate change. This need is particularly strong in mountainous catchments, which are the major source of streamflow for

hydropower production, and are particularly sensitive to climate change (Schaefli et al., 2007; Zierl and Bugmann, 2005). Energy companies across Switzerland are renewing and renegotiating their water concessions, transforming their existing infrastructure, and considering investments in new regions and energy sectors (Barry et al., 2015; SWV, 2012). However, in the vast majority of these cases, tailored analyses of climate change impacts are not used (Tonka, 2015).

To anticipate climate change impacts on hydropower production and to develop adaptation strategies, it is essential to account for end-user vulnerabilities and hydroclimatic changes at the local scale (Schaefli, 2015). Currently, the majority of studies that perform a climate change impact analysis focus on the effect of climate change on the seasonal cycle or on extreme events (Etter et al., 2017; Finger et al., 2012; FOEN, 2012; Hänggi and Weingartner, 2012; Köplin et al., 2014; Lopez et al., 2009; Vano et al., 2010), but rarely a combination of both. Furthermore, runoff quantities (water supply) are

often analyzed in isolation and not combined with climate projections (energy demand) related to hydropower management (Gaudard et al., 2013). However, in recent studies (Anghileri et al., 2018; Gaudard et al., 2018b; Savelsberg et al., 2018) modeling of the electricity market was combined with hydrological simulations for climate change to project potential revenue changes. These studies contribute to bridging the gap between economists and hydrologists and account for the interconnected nature of water and energy, which is fundamental for sustainable hydropower development. Yet, similarly to

the previously mentioned papers, these studies tend to focus on the seasonal cycle. A detailed overview of recent research on the impact of climate change on hydropower is provided by Savelsberg et al., (2018). The focus on particular runoff quantities is often determined by what climate and hydrological modelers perceive as most adequate and relevant (an approach commonly referred to as "top-down", see Wilby, Robert L.; Dessai 2010). However, this does not necessarily correspond to the needs of decision-makers in charge of designing adaptation strategies. Top-down studies typically provide

an overview of the impacts of climate change on hydrological resources, yet, for decision makers to assess the future profitability of their operations, more specific and local information is often needed (Vano et al., 2018). Given the costs associated with addressing and preparing for climate change impacts, it is essential to minimize the risk of maladaptation (i.e. inappropriate reaction to a threat) resulting from a misunderstanding of end users' vulnerabilities to climate change or from ill-designed projections (Broderick et al., 2019). Robust adaptation measures that provide benefits under a range of

climate change scenarios are especially valuable, given the reduced chance for maladaptation. Prioritizing stakeholder involvement early on (i.e. following a so-called "bottom-up" approach) enables the stakeholders to expose their concerns regarding climate change and to establish which potential future changes should be assessed in priority. This leads to





adaptation strategies, which are both autonomous (i.e. can be performed locally; IPCC 2008) but also dynamic and multi-sectoral (e.g. evolving while considering uncertainties related to modeling, economic, and social aspects; IPCC, 2012).


Here we present a case study relying on a bottom-up approach for creating hydrological and climatological projections tailored to support hydropower climate change adaptation. We collaborated with Groupe E, a Swiss electricity company that manages and has shares in several hydropower reservoirs in Switzerland. This project started with meetings with Groupe E managers, thereby involving them in the design of the study from the beginning. We relied on their expertise and asked them

to identify which hydroclimatic changes their business is most vulnerable to and to indicate change thresholds beyond which their activities would be significantly impacted. These meetings enabled us to pinpoint vulnerabilities of Groupe E's operations to climate change and to select hydrological indices and energy demand indices to characterize the associated risks. Furthermore, the figures showing our simulations were subjected to multiple rounds of feedback with Groupe E, and were tuned to enable Groupe E to discern future opportunities and risks related to their operations. Hence, over the course of

this study, we addressed the following research questions:

1. What are the vulnerabilities of Groupe E's operations to climate change?
2. How will inflows to Groupe E's reservoirs be altered by climate change?
3. What are the most considerable risks and opportunities induced by climate change and how may Groupe E adapt to
these changes?

This paper is organized in the following way: Section 2 introduces the Groupe E, the installations considered for this project, and it describes the indices and associated thresholds selected by Groupe E. Section 3 describes the observational and modelled data as well as the modeling framework employed to carry out hydrological (water supply) and climatological

(energy demand) projections. Section 4 presents the projected changes of the indices chosen by Groupe E. Section 5 discusses the implications of these changes for Groupe E's operations, and discusses possible future extensions of this study. In section 6, we summarize our results and draw conclusions regarding the use of bottom-up approaches in climate change impact analyses.

## 2 Project scope and identification of Groupe E's vulnerabilities to climate change

**2.1 Hydropower company and study catchments**

For this study, we interacted with two Groupe E energy managers and provided them with projections of the inflow entering two of their reservoirs. Groupe E is headquartered in Granges-Paccot in the Canton of Fribourg. Considering all of Groupe E's installations and purchases from the energy market, the company distributes an average of 2451 GWh/year to nearly 400,000 inhabitants. The company's electricity generation fleet consists of 6 dams and 10 power stations. The installations





are located either directly along the Sarine River or on one of its tributaries. Groupe E produces 1329 GWh of energy yearly, which is approximately 35% of the energy that they distribute. The remaining 65% is balanced by purchasing and trading on the energy market.

This study focuses on two of Groupe E's reservoirs: (i) the Vernex (Rossinière) dam - Montbovon power station and (ii) the
Montsalvens dam - Broc power station (Figure 1). The catchments of Montsalvens and Vernex have areas of 172.7 and 398.5 km$^2$, respectively (Table 1). The Vernex and Montsalvens installations are situated upstream of several other installations belonging to Groupe E, which turbinate water from the Sarine river along its lower reach. Given the placement of the Montsalvens and Vernex installations, their future functionality and security are crucial for Groupe E. We explored the future inflow into these two reservoirs in order to support adaptation to climate change, and in particular, the negotiation of a
new water concession for the two basins, as discussed in Section 2.1.1. Groupe E is familiar with ensembles and uncertainties associated with hydrological simulations, as they use ensembles of short-term hydrological forecasts for their daily operations.

### 2.1.1 Negotiations of the water concessions

In Switzerland, the sovereignty of public waters is assigned to the cantonal or local/municipal authorities, which can grant the right to use water for electricity production to a hydropower company via a lease known as a concession (Mauch and Reynard, 2004). Most dams in Switzerland were built between 1945 and 1970, and water concessions were then typically granted for a maximum of 80 years. Therefore, many energy managers are currently faced with challenges spurred on by the ending of their water concessions (SWV, 2012). Lac du Vernex is a reservoir with concession agreements with the cantons
of Vaud and Fribourg, which both end in 2052. Lac de Montsalvens is a reservoir located in the canton of Fribourg, and has a concession agreement with the canton of Fribourg ending in the year 2076. Typically, the submission for renewal is due 15 years in advance (i.e. the submission for renewal is due in 2037 for Vernex and 2061 for Montsalvens). Given the liberalization of the Swiss electricity market, new competitors are entering previously closed markets. Therefore, some hydropower companies may consider the early renewal of their concessions decades in advance in order to ensure their
production portfolio and to position themselves securely in the market. In addition, projections of climate change on relevant streamflow indices offer Groupe E insight into their resource availability in the future, and also help them gauge the flexibility of future operations. Therefore, Groupe E and other hydropower companies are currently interested in obtaining projections of future impacts of climate change, although the renewal of their concession is in theory only due in a few decades.

During concession negotiations, Groupe E representatives stated that the following would be considered (i) the development of the energy market and competitors, (ii) the projected supply of water resources, (iii) changes in energy demand, and (iv)


costs associated with adhering to new environmental standards. This study focuses on the estimation of future water

resources (point ii) and providing preliminary insights into future energy demand (point iii). During concession negotiations,

the respective authorities and Groupe E will agree upon the duration of the contract and the terms of the water fee (i.e., the

price to be paid by Groupe E to the owner of the water right). The water fee is determined based on the gross capacity of the

hydropower plant and elevation differential (head), as well as the amount of water that can be used for electricity production

under particular hydrological conditions as defined in the concession (Betz et al., 2019). Related to point (iv), a key aspect in

the negotiations of a water concession are new environmental regulations that hydropower companies now must comply

with, such as new residual water flow requirements. Environmental impacts on the ecosystem were not a primary concern in

the early stages of hydropower in Switzerland (Tonka, 2015). However, it is now well understood that hydropower systems

impact the natural connectivity, temperature, and dynamics of rivers and therefore have substantial impacts on the

downstream ecosystem (e.g. fish habitat). Swiss environmental regulations are listed within the Water Protection Act

(Gewässerschutzgesetz), which sets the rules for residual water flow, and defines residual flow as the amount of water that

must remain in a river after water withdrawals. Cantonal requirements are currently being strengthened to increase the

amount of residual flow required to remain in streams, which reduces the amount of water for hydropower production

(discussed further in Section 5.1.2).

### 2.1.2 Vulnerabilities to climate change and selection of indices and thresholds

Our discussions with Groupe E representatives enabled us to identify three main types of vulnerabilities: (i) water volume

vulnerabilities (will seasonal changes in inflow distribution impact the reservoir profitability, given that energy prices have

historically been highest in winter because of the high electricity demand?), (ii) low flow vulnerabilities (will low flow

situations become more frequent and make it more challenging to guarantee a residual discharge?), and (iii) high flow

vulnerabilities (will high flow situations become more frequent and how may they be used for profit?). To specifically

address these vulnerabilities, streamflow indices were selected together with Group E along with thresholds whose

exceedance would significantly impact Groupe E's production activities and profit. These hydrological indices and their

relevance for Groupe E operations are summarized in Table 2. While future changes in the mean monthly streamflow cycle

have been well explored (Addor et al., 2014; Smiatek et al., 2012; Vicuna and Dracup, 2007; Zierl and Bugmann, 2005),

studies focusing on changes in other streamflow characteristics, such as extremes (Köplin et al., 2014), are less common.

Groupe E representatives stated that although changes in the long-term mean monthly cycle are crucial, additional

hydrological indices are necessary to inform their concession negotiations and adaptation efforts.

Aside from hydrological indices, Groupe E also requested an assessment of the rain versus snow contribution to runoff so

that they can gain insight into their seasonal-scale operations. Historically, the Vernex and Montsalvens reservoirs reach

their highest level in May after the spring runoff. The onset of the convective storm season is around May/June as well. The

coincidence of meltwater and high intensity precipitation events can therefore lead to excess storm flow entering the





reservoirs, which must be released without turbination resulting in a profit loss and possible damage downstream. We used a hydrological model to characterize the respective contribution of rain and snowmelt to discharge (see Section 3.3.1).

Finally, two indices were chosen by Groupe E to gain insights into future energy demand: cooling degree days ($C_{DD}$) and heating degree days ($H_{DD}$). They were computed following the method presented in Gaudard et al., (2013), and is solely based on air temperature as shown in Eq. (1) and (2):

$$H_{DD} = \max \sum (Th - \theta t, 0) , \tag{1}$$
$$C_{DD} = \max \sum (\theta t - Tc, 0) , \tag{2}$$

where $\theta t$ is the air temperature retrieved from climate projections (Sections 3.2.2 and 3.2.3). The thresholds Th= 13°C and
Tc = 18.3°C were provided by Groupe E and correspond to the threshold values used in Gaudard et al., (2013). They represent the air temperatures that, when reached, cause consumers to turn on either cooling or heating in their homes. $C_{DD}$ and $H_{DD}$ were calculated for the cities (canton boundaries) of Zurich and Geneva, given that these areas comprise of typical Groupe E energy consumers. Results for Geneva are shown below and results for Zurich can be found in the Supplementary Materials-S8.

**3 Data and methods for impact modeling**

**3.1 Modeling framework**

To assess future changes in the streamflow and energy demand indices introduced above, we relied on the following model chain. We combined two greenhouse gas emission scenarios (see Section 3.2.1), eleven regional climate models forced by general circulation models (GCM-RCMs; see Section 3.2.2), two GCM-RCM post processing methods (see Section 3.2.4)
and one hydrological model to simulate inflow entering the two reservoirs (Figure 1). The hydrological model was calibrated using three objective functions and ten optimized parameter sets were generated per objective function and per calibration period (see Section 3.3.3). This modeling framework follows the procedure outlined in Hakala et al., (2019). It enabled us to assess uncertainties in the projected discharge and to provide Groupe E with a projected likely range for each index under future climate. The following subsections describe the steps of our modeling chain in greater detail.

**3.2 Climate data and preparation**

**3.2.1 Emission scenarios**

Representative concentration pathways (RCPs) are scenarios describing possible futures for the evolution of Earth's atmospheric composition, and as such, provide boundary conditions for climate models. RCPs 4.5 and 8.5 were selected for this study. RCP 4.5 corresponds to an intermediate emission trajectory, where greenhouse gas (GHG) emissions peak around



2040 and then generally stabilize. In contrast, RCP 8.5 assumes that GHG emissions will continue to increase throughout the 21st century (Meinshausen et al., 2011).

### 3.2.2 Observational and GCM-RCM data

Observational meteorological data were retrieved from the 2 km MeteoSwiss gridded datasets of TabsD (Frei, 2013) and RhiresD (Frei and Schär, 1998; Schwarb, 2000). The daily reservoir inflow was estimated by Groupe E for the period of

2008-2018 by solving the water balance based on variations of the reservoir level, the volume of water turbinated for hydropower production and estimated losses due to evaporation from the reservoir (reservoir losses to the groundwater were neglected). GCM-RCM temperature and precipitation data were retrieved from the Coordinated Regional Downscaling Experiment for Europe (EURO-CORDEX; http://www.euro-cordex.net/, see Table 3). GCM-RCM model selection followed the methodology described in Hakala et al., (2018), which entails selecting models based on their hydrological performance

over the historical period. Furthermore, we excluded models generating snow towers because of the influence that cooler temperatures associated with the snow towers may have on the climate change signal (Frei et al., 2018; Hakala et al., 2018; Zubler et al., 2016). EURO-CORDEX provides simulations at both 0.44° and 0.11° resolution, but only 0.11° data was used given the size of the catchments investigated in this study. Overall, the exclusion of some GCM-RCMs due to their poor hydrological performance results in a tailored modeling setup that prioritizes end-user decision making.

### 3.2.3 Data extraction

To extract temperature and precipitation from the gridded datasets, an area weighted method, as shown in Hakala et al. (2018), was used. As a first step, the grid cells of the meteorological data were overlaid with the shapefile of a given catchment. Once the data from the overlapping grid cells were extracted, a weight was applied to each grid-cell time series based on the percentage of catchment area overlapped by the grid cell, resulting in a single catchment-mean time series. This

area-weighted methodology was used to extract temperature and precipitation data from both the EURO-CORDEX and MeteoSwiss datasets. In the case of the EURO-CORDEX dataset (horizontal grid spacing of ~12.5 km), nine grid cells at least partially overlapped with the Vernex catchment and four grid cells overlapped with the Montsalvens catchment.

### 3.2.4 Bias correction

The GCM-RCM simulated temperature (T) and precipitation (P) time series were bias corrected using a nonparametric

quantile transformation of seasonal distributions. The cumulative distribution functions (CDFs) were determined individually for the different seasons: December-February (DJF), March-May (MAM), June-August (JJA), and September-November (SON) for both the observed (MeteoSwiss) and simulated (EURO-CORDEX) T and P time series. For GCM-RCMs with a non-leap-year calendar (Table 3), T and P were converted to a Gregorian calendar prior to bias correction. For GCM-RCMs with a 360-day calendar, observational data were converted to a 360-day calendar before bias correction and

the hydrological model was run using this calendar. The 'qmap' package in R (Gudmundsson, 2016; Gudmundsson et al.,





2012) was used to match the CDF of the simulated data to that of the observed data. Specifically, a transfer function was generated to match each raw GCM-RCM P and T percentile to the associated P and T percentile of the MeteoSwiss data. The biases in the raw GCM-RCM simulations were assumed to be stationary over time, so the same transfer functions were used to correct the projections of T and P.

## 3.3 Hydrological data and model

### 3.3.1 Hydrological model

The bucket-type Hydrologiska Byråns Vattenbalansavdelning (HBV) model (Bergström, 1976; Lindström et al., 1997) was used to simulate streamflow entering the two reservoirs. For this project, we used the version HBV-Light (Seibert and Vis, 2012). HBV is a semi-distributed model that uses four routines (snow, soil, response, and routing routines) and relies on elevation bands to account for changes in T and P with elevation within a catchment. HBV requires temperature, precipitation, and potential evaporation time series as input. For a more detailed description of the separate routines, we refer the reader to Seibert and Vis, (2012). For the remainder of the manuscript, we use the term HBV when referring to the version HBV-Light.

### 3.3.2 Adjustment of discharge data

When initially analyzing the discharge data provided by Groupe E in combination with MeteoSwiss observational meteorological data, we noticed that precipitation was too small to explain the discharge flowing into the Montsalvens reservoir. Based on water balance calculations informed by karst hydrogeological information (Bitterli et al., 2004) and actual evaporation estimates (Menzel et al., 1999), it was assumed that karst was responsible for the larger than expected discharge. The Montsalvens and Vernex catchments are located in a transitional region between the Alps and the Swiss Plateau. As pointed out by Fan, (2019), a catchment is more likely to be an open or 'leaky' system when positioned at either the high or low end of a steep regional topographic and climate gradient, which is the case here. Therefore, a correction factor was applied to the observed discharge to re-scale it to match the expected mean discharge. The factor was calculated following the water balance equation: $P = E + (f \cdot Q) + \Delta S$ for the period 2008-2018, where P represents precipitation falling within the catchment, E stands for actual evaporation, Q represents the inflow reported to enter the Montsalvens reservoir, and $\Delta S$ stands for change in storage, which was considered negligible in this case. By applying the factor f (0.79) to the discharge time series, we were able to close the water balance equation. This method therefore assumes that 21% of the total inflow entering the Montsalvens reservoir is groundwater entering through the karst system. Karst hydrogeology did not appear to have a discernible effect on discharge for the Vernex catchment.





### 3.3.3 Calibration and validation

Calibration and validation of HBV was based on three different objective functions, namely the Lindström measure (Lindström et al., 1997), Nash-Sutcliffe efficiency (Nash and Sutcliffe, 1970), and Kling-Gupta efficiency (Gupta et al., 2009). Two separate periods were used for calibration and validation: 01-10-2008 to 30-09-2013 and 01-10-2013 to 31-08-2018. For each combination of objective function and time period, ten independent parameter sets were generated. HBV was calibrated using a genetic algorithm and Powell optimization (Seibert, 2000) method (10 000 model runs for the genetic

algorithm and an additional 1000 runs for the Powell optimization). Using multiple objective functions and calibration periods enabled us to account for parameter uncertainty and to generate an ensemble of equally likely realities (Brigode et al., 2013; Coron et al., 2012; Klemeš, 1986). Both catchments achieved reasonable calibration and validation scores (above 0.75 NSE or higher for all objective functions and periods). Therefore, all parameter sets were carried forward in the modeling chain.

### 3.4 Evaluation of the modeling chain over the reference period

Prior to creating projections, we analyzed our modeling chain performance over a reference period. Figure 2 provides a comparison between (variable)$_{obs}$ and (variable)$_{ref}$ for each hydrological index and climate change impact index. The ref subscript indicates that the index was computed using HBV simulations driven by observed atmospheric forcing. In the case of the hydrological indices, $Q_{obs}$ and $Q_{ref}$ stem from different time periods, as Group E records only cover the period 2008-

2018. Given this mismatch in time periods, we began by comparing the monthly precipitation of the $Q_{obs}$ and $Q_{ref}$ time periods (Supplementary Materials-S1). The period 1980-2009 ($Q_{ref}$ period) experienced a wetter climate than 2008-2018 ($Q_{obs}$ period).

Figure 2 shows that hydrological simulations driven by raw climate simulations present severe biases. For instance, the mean

monthly inflow is vastly overestimated by raw data from April through December (Figure 2a). Bias-correction leads to a significant reduction of these biases, and in fact, was necessary here to capture the indices required by Group E (Figure 2a to 2h). Figure 2g shows that the application of bias correction is successful at reducing the ensemble spread of $H_{DD\ raw}$ (yellow shaded area), resulting in $H_{DD\ qm}$ (purple shaded area). $H_{DD\ qm}$ can be seen to fit well with $H_{DD\ ref}$ for the entirety of the annual cycle. Figure 2h also shows a reduction of the $C_{DD\ raw}$ ensemble spread (yellow shaded area) due to the application of

quantile mapping ($C_{DD\ qm}$; purple shaded area), with August retaining a relatively high level of uncertainty. As Groupe E concession negotiations require more finely tuned projections than what can be delivered by raw simulations, we excluded simulations generated using raw GCM-RCM data from the results section, so that the focus can be on future changes and not on the effects of the bias-correction. Figures displaying hydrological variables utilize two y-axes where specific discharge (mm/day) is shown on the left-hand axis, and the discharge (m³/day) is displayed on the right-hand axis. The former allows



for a comparison between catchments, whereas the latter is more useful for energy managers, such as Groupe E, when operations primarily are looked at in terms of volumes.

Overall, when using bias-corrected climate simulations, HBV satisfactorily captures the annual discharge cycle (Figure 2a), the respective contribution of snow and rain to streamflow (Figure 2b) as well as Q5 and Q95 during the seasons of interest

(Figures 2c and 2d). In contrast, HBV tends to overestimate both the duration of periods below Q5 and above Q95 (Figure 2e and 2f). It is however, important to note that HBV was not specifically calibrated against the hydrological indices mentioned in Table 1, and so it is not surprising if $Q_{obs}$ and $Q_{ref}$ deviate when compared across these indices. Pool et al., (2017) showed that HBV tends to underestimate streamflow characteristics related to mean and high flow conditions and generally overestimates low flow conditions. For this study, it is the relative change that is most important for Groupe E. A

comparison of the $Q_{obs}$ and $Q_{ref}$ inflow time series (Supplementary Materials-S2), plotted over their shared period (2008-2010), shows that HBV generally underestimates low flows, which is consistent with the results shown in (Pool et al., 2017).

### 3.5 Projections of climate change impacts

Since the performance of the modeling chain was considered to be satisfactory over the reference period, all parameter sets generated in Section 3.3.3 were used to simulate projections for the periods of 2020-2049, 2045-2074, and 2070-2099. Our

modeling chain comprised of: two emission scenarios (RCP4.5 and RCP8.5), eleven EURO-CORDEX GCM-RCMs, two post processing methods (raw and quantile mapping), one hydrological model (HBV), three objective functions for the hydrological model (Lindström measure, Nash-Sutcliffe efficiency, Kling-Gupta efficiency), along with ten optimized parameter sets per objective function, and two calibration periods. This lead to a total of 1320 bias corrected simulations for each future period and basin. Below we focus on the comparison between 1980-2009 and 2070-2099 under RCP8.5, as it is

the most interesting for Group E, and on the Vernex catchment. The results and figures for all periods, RCP4.5 and both catchments were provided to Groupe E and the end-of century results for Montsalvens can be found in the Supplementary Materials. The projected streamflow indices were not compared to observed discharge data, because such a comparison could be misleading due to the mismatch in time periods and the inclusion of hydrological model uncertainty. Instead, the projections were compared to simulations for the reference period based on bias-corrected GCM-RCM simulations.

## 4 Results

### 4.1 Water volume

Figure 3a compares historical (1980-2009) and future (2070-2099) annual distribution of inflow entering the Vernex reservoir for RCP 8.5. Winter (DJF) discharge is shown to widely exceed the +20% and +50% thresholds specified by Groupe E. Meanwhile, summer (JJA) discharge decrease is expected to be around the -50% threshold. Groupe E asked for

the long-term mean monthly discharge cycle to be visualized by showing the volume difference between future (2070-2099)





and historical (1980-2009) conditions. Figure 3b was requested so that the total amount of water gained/lost can be directly considered during concession negotiations. Under RCP 8.5, the Vernex reservoir will experience more inflow between December and March, but less inflow from May to October. By the end of the century, the expected average change in inflow for the Vernex reservoir is -1.11 M m³/day (-4.52 to +2.54 likely range) under RCP 8.5 and -0.24 M m³/day (-2.968 to +2.3487 likely range) under RCP 4.5. Similarly, the inflow entering the Montsalvens reservoir will experience an average decrease of -0.724 M m³/day (-2.19 to +0.81 likely range) under RCP 8.5 and -0.18 M m³/day (-1.61 to + 1.08 likely range) under RCP 4.5.

The shift in the annual distribution of inflow entering Groupe E's reservoirs is primarily caused by changes in the form of precipitation contributing to inflow (Figure 4). Peak annual contribution to inflow from snowpack is expected to decrease by more than half and to occur earlier in the year, shifting from May to April. Spring runoff derived from snowpack will be a less reliable source of inflow in the future. Meanwhile, rain will decrease its respective contribution to inflow over the summer. The shift in spring runoff and the reduction of rainfall contribution to inflow results in a reduction of inflow entering Groupe E's reservoirs (Figures 3b, S4). Over the 21st century, winters will see an increasing rain contribution to inflow, and a reduced contribution from both rain and snow from May until November. The Montsalvens catchment is expected to experience a similar regime change in the future, with an even more pronounced reduction of snowfall contribution (Supplementary Materials-S5).

## 4.2 Low flows

$Q_{qm}$ simulations of low flows (Q5) for JJA and SON strongly decrease under RCP 8.5, with the majority of the ensemble indicating a decrease greater than the -50% threshold (Figure 5a). The spread of the ensemble for both seasons is relatively small in absolute terms. Projections for the inflow entering the Montsalvens reservoir indicate similar changes, with Q5 dropping below the -50% threshold for JJA and the median of the SON ensemble lying close to the -50% threshold (Supplementary Materials-S6).

The frequency of consecutive days below Q5 is expected to increase under the influence of climate change in SON. Figure 6a demonstrates this concept by showing the cumulative distribution functions (CDFs) of the consecutive days below Q5 for the Vernex catchment over the SON season. The robust nature of the change compared to historical simulations demonstrates that there is high confidence that there will be more days below Q5 over the SON season in the future, although it should be noted that $Q_{qm}$ data initially overestimated the CDFs of consecutive days below Q5 (Figure 2e). The results for the Montsalvens reservoir agree with the changes shown for the Vernex reservoir, with a slightly less pronounced difference between the historical and future periods. For the Montsalvens catchment, there are relatively fewer extended periods of low flow (Supplementary Materials-S7).



### 4.3 High flows

The magnitude of high flows (Q95) is expected to decrease in JJA under RCP 8.5 (Figure 5b). However, the median and the
majority of ensemble members are within the 50% interval designated by Groupe E. In contrast, for winter, $Q_{qm}$ simulations
show a significant increase, far exceeding the +50% threshold specified by Groupe E. Inflows entering the Montsalvens
reservoir exhibit similar behavior over both seasons (Supplementary Materials-S6).

More extended periods of consecutive days above Q95 are projected in DJF under the influence of climate change. The
CDFs of the future simulations show a significant increase in the length of consecutive high flow periods, including periods
longer than the stipulated ten-day threshold. Results for Montsalvens indicate similar but less pronounced changes
(Supplementary Materials-S7).

### 4.4 Energy demand

Figure 7a shows that the number of HDD will decrease over the winter months under the influence of climate change,
whereas the summer months experience no change given that this time of year is already too hot to invoke heating within a
household. Figure 7b shows that $C_{DD}$ will increase for the months between May and October. The winter months show no
change given that these months are too cold to invoke cooling within the household of a typical Groupe E customer.
Projections of the Canton of Zurich show a general agreement with the magnitude and distribution of change
(Supplementary Materials-S8).

## 5 Discussion

### 5.1 Climate risks and opportunities and possible adaptation strategies

### 5.1.1 Water volume

Of the selected indices, Groupe E stated that a seasonal breakdown of changes to water volume are the most important
indices for their operations. Some changes in the seasonal inflow distribution represent new opportunities for Groupe E.
Over the winter period, the Lac du Vernex will experience an average increase of +90% under RCP 8.5 and +63% under
RCP 4.5. Lac de Montsalvens will experience an average increase of +89% under RCP 8.5 and +61% increase under RCP
4.5. Hydropower will likely remain an important supplier of electricity in the winter given the low yield of photovoltaics
during the short winter days and the unpredictability and contentious politics of wind power (Kienast et al., 2017). Therefore,
these changes could allow Groupe E to capitalize on generally higher energy prices in winter, resulting in a potential increase
in profits for this season.





In contrast to the winter, the regime changes in the summer and fall are expected to lead to new challenges for Groupe E. Over the summer period, the Lac du Vernex will experience an average decrease of -51% under RCP 8.5 and -30% under RCP 4.5. Lac de Montsalvens will experience an average decrease of -49% under RCP 8.5 and -28% decrease under RCP

4.5. The reduction of summer inflow can be linked to the snowpack shrinkage over the coming century and the simultaneous reduction of total precipitation over the summer months (Figure 4). Köplin et al., (2014) showed that when snow accumulation is important to a catchment hydrological regime during the historical period, the anticipated changes in seasonality are most pronounced. Groupe E stated that the Vernex and Montsalvens reservoirs are too small to store water over the winter period in order for it to be used to offset droughts in the summer period. Adjusting the size of their reservoirs

is currently not a viable option and therefore it was not necessary to adjust the size of the reservoir during modeling exercises. Given a decrease in inflow over the summer, and a likely increase in energy demand for cooling (Figure 7b), Groupe E may need to consider increasing their investments into other energy sectors such as photovoltaics, which correlates nicely to the relatively longer daylight hours in summer.

Groupe E stated that in addition to other market conditions and legal requirements, they could use this information to negotiate a lower cost for their water fee, which is in part determined based on the amount of water that can be used for electricity production under particular hydrological conditions. However, the water fee framework is subject to re-review in 2024 (Betz et al., 2019). For an impact comparison of the different water fee systems on Swiss hydropower, see Gaudard et al., (2018a).

**5.1.2 Low flows**

Low flows will require special attention in the coming decades, as the magnitude of Q5 will reduce drastically compared to historical values, with the majority of ensemble members exceeding the -50% threshold in JJA and SON (Figure 8a & b). In addition, droughts will increasingly extend beyond Groupe E's 60-day threshold in JJA and SON (Figure 8a & b). Groupe E stated that these changes would likely influence the negotiated terms of the water fee. The decrease in production over a long

period of time has a significant effect on the flexibility of production. Flexibility is a significant component of a storage hydropower plant's profitability, as it enables water managers to turbinate when electricity prices are optimal. Under climate change, as flexibility decreases and energy demand likely increases due to heat waves (Section 5.1.4), Groupe E stated that they would consider acquiring new sources of energy production to compensate for this loss.

Cantonal requirements are currently being strengthened to reduce environmental impacts. One of the cantonal measures include increasing the amount of residual flow for environmental reasons (e.g. flora, fauna, and sediment transport are affected by very low flows). However, the results of this study demonstrate that the water carried by low flows will substantially decrease over the coming decades and the duration of low flow conditions will increase. Therefore, during





concession negotiations, Groupe E may request that residual flow requirements should not increase or find a middle ground
given the future behavior of low flows entering their reservoirs.

### 5.1.3 High flows

Opportunities are present over the winter period, as the average high inflows to the Vernex and Montsalvens reservoirs are
projected to increase by more than 50% (Figure 8d) and exceed the ten-day threshold specified by Groupe E (Figure 6b). An
increase of high flows entering the reservoirs during the winter period, when energy prices are highest, would allow Groupe
E to better satisfy demand using their own production, rather than supplementing their supply by trading/purchasing on the
energy market. Groupe E stated that this increase is generally seen as a positive outcome for their operations given that
energy prices are usually highest during the winter. The hydrological shift from slow, snow-dominated processes to more
variable, rainfall-driven processes will require Groupe E to adapt the management of their reservoirs so that these quick
inflows can lead to increased profit. This would require Groupe E to capture and turbinate the inflow at optimal times or at
pre-arranged prices. Conversely, projections show a decrease in high flows in the summer (Figure 8c), which indicates a
reduced risk of water loss due to spillovers. Groupe E could consider investing in their existing short-term forecasting and
trading unit in order to improve their forecasts of high flow events. As Groupe E can decide when to sell its electricity
anytime between the next hour to the next 3 years, a balance between best price and risk management needs to be found.

### 5.1.4 Energy demand

Groupe E will need to supply their consumer's increasing energy demand during the summer and fall seasons (Figure 7b),
which will be challenging if only inflow entering their reservoirs is considered. Ownership in other energy sectors may help
offset losses directly associated with hydropower production. The Swiss Energy Strategy 2050 stipulates that the energy
deficit left from the decommissioning of nuclear energy should be partially replaced with an increase in hydropower
production. However, Switzerland has almost reached its maximum capacity for hydropower production, and projections of
future energy mix show that renewables (e.g. wind and photovoltaics) will need to play a significant role in supplementing
the energy deficit left by the phase out of nuclear energy (Redondo and Van Vliet, 2015).

Electricity companies that hope to adapt to climate change cannot base their strategies on water availability alone (Gaudard
et al., 2013; Savelsberg et al., 2018). This motivated the selection of the energy demand index that was chosen for this study.
Although a temperature-based energy demand approach is inherently limited, alternative projections of how the electricity
market will develop over time are highly uncertain and only a limited amount of research has been done to assess the impact
of energy and economic policies on future hydropower (Anghileri et al., 2018). Nonetheless, if operations can be adapted to
an expected increase in electricity price volatility, a more flexible operating strategy could allow Groupe E to capitalize on
peak prices (see Section 5.4 for further discussion).



### 5.2 Benefits of developing tailored projections by following a bottom-up approach

Following a bottom-up approach and involving stakeholders in the modeling and figure design provided key benefits and insights. It revealed, for instance, that the indices often chosen by impact modelers are not necessarily well-suited to support decision-making. Although these standard indices, such as the long-term mean monthly distribution of inflow are useful, interacting with Groupe E made clear that, given the complexity of their decision-making process, a singular index or non-tailored indices are of limited use. This led to the selection of a less common index, consecutive days of low flows, which enabled us to explore a critical vulnerability for hydropower operations often overlooked by top-down impact studies. The importance of tailored projections and indices is especially apparent when the projections presented here are compared to existing literature on climate change impacts on hydropower production. The mean monthly inflow changes that the Vernex and Montsalvens catchments will experience are most comparable to projections shown for the nearby Emme catchment simulated by (Addor et al., 2014). But given the local-scale information needed for hydropower management and concession negotiation, indices beyond the long-term mean monthly cycle are needed. (Finger et al., 2012) produced hydrological projections for the Saas Fee region in Switzerland, but these are not directly useable by Groupe E, as the hydrological indices they analyzed are not specific enough for concession negotiations nor is the alpine region they cover expected to respond in the same way to climate change as region of Groupe E's catchments.

Groupe E managers expressed that our collaboration enabled them to see the impacts of climate change at the local level, in a way that allows them to envision the impacts they may experience on a daily basis as energy managers. Groupe E is interested in similar studies for other catchments and they mentioned that they would consider investing in new hydrological projections as models develop over time. Importantly, they stressed that it is useful to have the projections as early as possible in order to begin the process of climate change adaptation and to prepare for critical conversations prior to official negotiations. They stated that this collaboration made the climate change phenomenon much more real and that, compared to generic information they have access to, the figures we co-produced provide them with a particularly clear picture of the likely impacts of climate change on their activities. This highlights the benefits of the direct inclusion of stakeholders to anticipate and efficiently prepare for future climate change impacts.

### 5.3 Visualizing climate change impact projections and their uncertainties to inform decision-making

The design of all the visuals presented in this study was informed by Groupe E. A decision-analytic summary figure was created based on Figure 2 in (Brown et al., 2012) and was proposed initially to Groupe E. This type of figure uses two axes to represent changes in two selected variables and indicates which decision is optimal for different regions of this two-dimensional space. Groupe E pointed out that, given their situation, this type of visual is limited as it is too simple to display the numerous considerations influencing the concession renewal. Instead, Figure 5 in Broderick et al., (2019) was used as a





basis for Figure 8 to succinctly visualize changes in a series of key indices in relation to thresholds prescribed by the Groupe E. A summary table of the main opportunities and adaptation options was also provided to Groupe E (Table 4).

Characterizing and visualizing projections uncertainty also played a central role during this project, as Groupe E must negotiate their water concessions despite an abundance of uncertainty. We hence made sure to understand how Groupe E interprets uncertainty bands associated with the projections of their water resources. Uncertainties associated with model calibration and validation and multi-model ensembles were already familiar to Groupe E because they routinely utilize these methods and incorporate these uncertainties into their day-to-day operations. They recognize the value of accounting for uncertainties, in particular as this helps to prevent maladaptation, as deterministic projections convey an unjustified

impression of certainty. Groupe E explained that they consciously consider the width of uncertainty bands compared to the mean change in order to render confidence in the figure. For instance, Figure 5a shows the magnitude of Q5 over the JJA and SON seasons between the historical (1980-2009) and future period (2070-2099). The spread of the projections is reflected by the width of the boxplots. Figures 5a shows a clear change between historical and future low flows, where all future ensemble members exceed the -50% threshold specified by Groupe E. This result represents a profit loss for Groupe E

because there will be less water available for turbination, and if turbinated, it will be at a lower efficiency. In other cases, when results are less definitive, Groupe E stated that the mean (or median) of the projections is most useful to them.

**5.4 Limitations and next steps**

This study focused on hydroclimatic changes using a range of streamflow indices. These changes are important but they only partially determine the profitability of the two reservoirs. These is now a need to complement this analysis with a more

economical analysis, focused on the future energy demand and on the evolution of the energy market. A collaboration between hydrological climate change impact modeling and energy-economic modeling seems to be the practical next step, e.g. Anghileri et al., (2018); Savelsberg et al., (2018). However, these studies tend to aggregate all climate change impacts by focusing solely on the profitability of the reservoir, and they tend to consider only changes in the seasonal cycle. This study shows how linking stakeholder vulnerabilities to changes in individual indices offers an approachable means to discuss

adaptation measures compared to a lumped profit/loss figure. A collaboration between hydrologists, economists, and relevant stakeholders would help to ensure the incorporation of all necessary components into a modeling framework to support concession negotiations and sustainable development of hydropower.

Additional streamflow indices would be useful to Groupe E, in particular related to magnitude and duration of flooding. Future

work could expand the hydrological indices to include rare and potentially damaging flooding events. The indices chosen by Groupe E should not be interpreted as universally appropriate for all hydropower climate change adaptation studies. Instead we advocate for stakeholder involvement early on so that indices, modeling chains, and results can be tailored for decision





making. Future work could also involve the characterization of sources of uncertainty not considered in this study, such as hydrological model uncertainty and natural variability.

## 6 Conclusions


This study illustrates the benefits of involving stakeholders in hydropower climate change impact studies, whereby decision-making was given top priority. This project went beyond a usual climate change impact assessment by addressing the specific needs and concerns of decision makers. This approach is essential given most hydrological climate change impact studies analyze runoff quantities in isolation but rarely address issues of water management (Gaudard et al., 2013; Hänggi and

Weingartner, 2012). This project was designed to provide guidance to Groupe E regarding their adaptation efforts to climate change and to support the negotiations of their water concessions. We asked Groupe E representatives to describe their main vulnerabilities to hydroclimatic variations, and together, we selected hydrological and energy demand indices to characterize future impacts. This enabled us to identify key opportunities and risks that are relevant to the concession negotiation process and climate change adaptation strategies.


The figures we provide will help Groupe E determine the value of water in the future and the price they are willing to pay for the renewal of their concessions. Our results indicate a significant increase of inflow over the winter period when energy prices have historically been at their highest. Alternatively, a reduction of summer inflows will present new risks for Groupe E, given an increase in energy demand for cooling due to increasing temperatures. Our projections of low flows provide a basis from

which Groupe E may negotiate for residual flow requirements to remain the same or only slightly increase. High flows represent an opportunity for Groupe over the winter, given that the winter period usually corresponds to higher energy prices. In order to generate profit, Groupe E will need to consider ways in which to store and turbinate these higher inflows. The involvement of Groupe E representatives early on in the project was vital towards ensuring that results and figures of this study were directly useful for their concession negotiations and to provide insights into how their operations are likely to be impacted

by climate change.

This case study demonstrates how projections of future climate change impacts on water resources can be tailored to support the negotiations for the re-licensure of water concessions. The environmental regulations that are agreed upon during the concession negotiations usually cannot be altered during the duration of the concession. Given the multi-decade length of a

concession, it is crucial for climate change to be considered at the onset of concession negotiations. The analysis presented here is transferable to other water management entities and this assessment provides guidance for other climate change projects that strive to follow a bottom-up approach and deliver projections that are directly useful for decision-making.



**Authors contribution:**

KH and NA designed this study based on previous exchanges between NA and Groupe E. KH refined the scope of the project with Groupe E over the course of several meetings. KH performed the climate change impact analysis, with oversight from NA and JS. Writing of the paper was led by KH with feedback from all coauthors. Finalization of the manuscript occurred primarily between KH and NA.

**Competing interests:**

The authors declare that they have no conflict of interest.

**Acknowledgements:**

This study was supported by the Swiss National Science Foundation via the grants 200020_156606 and P400P2_180791. The authors thank Marc Vis for support related to the HBV model. We also thank Prof. Dr. Hannes Weigt for providing useful insights into the Swiss water fee framework. We acknowledge the Federal Office of Meteorology and Climatology MeteoSwiss and the European Coordinated Regional Downscaling Experiment (EURO-CORDEX) for the atmospheric and climate modeled data, respectively.

**Supplementary Materials:**

Supplementary materials for this article are located as an attachment at the end. In addition, the following materials are available online for those who wish to carry out their own hydrological climate change impact analysis: (i) download of HBV hydrological model can be accessed: https://www.geo.uzh.ch/en/units/h2k/Services/HBV-Model.html, (ii) we refer the reader to Hakala et al., (2019), which is an encyclopedia chapter which introduces the steps of a hydrological climate change impact assessment.

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



**Table 1: Main characteristics of the two study catchments including catchment area, elevation, glacier coverage, karst percentage, forest cover, and energy production. Data for this table were derived from multiple sources: area and mean elevation of the catchment was provided by Groupe E and confirmed during delineation for modeling purposes, glacier coverage was estimated using satellite imagery from google, karst hydrogeology was estimated using a dataset provided by (Bitterli et al., 2004), and mean energy production was provided by the Swiss Federal Office of Energy (SFOE).**

| Reservoir- Dam | Area [km²] | Mean elevation [m a.s.l.] | Glacier coverage [%] | Karst hydrogeology [%] | Mean energy production [MWh/year] |
|---|---|---|---|---|---|
| Montsalvens- Broc | 172.7 | 1386 | 0 | 35 | 71 567 |
| Vernex- Montbovon | 398.5 | 1639 | <1 | 15 | 59 422 |



**Table 2: Hydrological indices selected after discussions with representatives of Groupe E. The relevance of each index for Groupe E's operations is explained, and vulnerability thresholds for each index are provided. Relative changes exceeding these thresholds would have a significant impact on Groupe E's operations. In cases where two thresholds are provided, the exceedance of the lower threshold represents a significant impact and the upper threshold represents a critical impact. Visual aids for each index are also provided in the far-right column.**


| Category | Hydrological index (season) | | Specific relevance for Groupe E | Vulnerability thresholds | Visual |
|---|---|---|---|---|---|
| Water volume | Long-term seasonal mean | March, April, May (MAM) | Snowmelt runoff may coincide with high intensity precipitation events that could overwhelm Groupe E's ability to store and turbinate reservoir inflows. | 20%, 50% | |
| | | June, July, August (JJA) | Meeting water demand for recreation, esthetic, and residual flow requirements. | | |
| | | September, October, November (SON) | Managing reservoir level given drought concerns, meeting demand for recreation, esthetic, and residual flow requirements. | 20%, 50% | |
| | | December, January, February (DJF) | Meeting energy demand during the coldest time of year, and identifying opportunities to benefit from historically high energy prices. | | |
| Low flows | Q5: 5th percentile of daily streamflow | June, July, August (JJA) | Meeting water demand for recreation, esthetic, and ecological purposes. Important for water fee negotiations and to assess whether regulations for residual flows are realistic. | 50% | |
| | | September, October, November (SON) | Meeting water demand for recreation, esthetic, and ecological purposes. Important for water fee negotiations and to assess whether regulations for residual flows are realistic. | | |
| | Consecutive days of low flow | June, July, August (JJA) | Help with storage management during extended drought, concern for meeting energy demand and residual flow requirements. | 60 days | |
| | | September, October, November (SON) | Storage management during extended drought, concern for meeting energy demand and residual flow requirements. | | |
| High flows | Q95: 95th percentile of daily streamflow | June, July, August (JJA) | Reservoir levels are at their highest, multi-day high intensity precipitation can lead to water release without turbination (profit loss) or damage downstream in extreme cases. | 50% | |
| | | December, January, February (DJF) | Explore opportunities to benefit from high energy prices during the winter months, although DJF has not produced as much inflow as JJA. | | |
| | Consecutive days of high flow | June, July, August (JJA) | Reservoir level management and balancing dam releases with high intensity precipitation events. | 10 days | |
| | | December, January, February (DJF) | Utilizing high volumes of inflow at times when the unit price of water is its highest. | | |



**Table 3: Overview of the eleven EURO-CORDEX GCM-RCM combinations used in this study. Some models were removed from the ensemble due to either snow tower issues or irregularities in the discharge simulations. The models which have been removed are denoted by light gray text and italicized font.**


| No | GCM | RCM | Calendar | Notes |
|----|-----|-----|----------|-------|
| 1 | CNRM-CERFACS-CNRM-CM5 | CLMcom-CCLM4-8-17 | Gregorian | |
| 2 | ICHEC-EC-EARTH | CLMcom-CCLM4-8-17 | Gregorian | |
| 3 | MOHC-HadGEM2-ES | CLMcom-CCLM4-8-17 | 360 | |
| 4 | MPI-M-MPI-ESM-LR | CLMcom-CCLM4-8-17 | Gregorian | |
| | *ICHEC-EC-EARTH* | *DMI-HIRHAM5* | *Gregorian* | *R-ST* |
| | *NCC-NorESM1-M* | *DMI-HIRHAM5* | *Gregorian* | *R-ST* |
| | *IPSL-IPSL-CM5A-MR* | *IPSL-INERIS-WRF331F* | *Gregorian* | *R-D* |
| | *ICHEC-EC-EARTH* | *KNMI-RACMO22E* | *Gregorian* | *R-ST* |
| | *ICHEC-EC-EARTH* | *KNMI-RACMO22E* | *360* | *R-ST* |
| 5 | MOHC-HadGEM2-ES | KNMI-RACMO22E | Gregorian | |
| 6 | MPI-M-MPI-ESM-LR | MPI-CSC-REMO2009 | Gregorian | |
| 7 | CNRM-CERFACS-CNRM-CM5 | SMHI-RCA4 | Gregorian | |
| 8 | ICHEC-EC-EARTH | SMHI-RCA4 | Gregorian | |
| 9 | IPSL-IPSL-CM5A-MR | SMHI-RCA4 | non-leap | C |
| 10 | MOHC-HadGEM2-ES | SMHI-RCA4 | 360 | |
| 11 | MPI-M-MPI-ESM-LR | SMHI-RCA4 | Gregorian | |

(R-ST): Removed due to snow towers in GCM-RCM model output
(R-D): Removed due to irregularities in the mean monthly distribution of discharge when simulations when forced by this GCM-RCM
(C): Calendar converted from non-leap year to proleptic Gregorian



**Table 4: The major opportunities and risks for Groupe E's operations diagrammed in relation to the hydrological and climatological considerations for concession renewal.**

| Vulnerability | Decrease in long-term mean monthly inflow | Increase in the duration of low flows | Seasonal change in the behavior of high flows | Meltwater mixing with rain events in spring could overwhelm Groupe E's operations | Decrease in energy demand |
|---|---|---|---|---|---|
| Index | • Seasonal means | • Q5<br>• Consecutive days below Q5 | • Q95<br>• Consecutive days above Q95 | • Rain versus snow contribution to inflow | • HDD<br>• CDD |
| Change in index | • Increase in long-term mean monthly inflow over the winter<br><br>• Decrease in long-term mean monthly inflow over summer and fall | • Magnitude of water carried by Q5 will reduce by 50% over the summer and fall<br><br>• The duration of low flows below Q5 will extend as long as 80-90 days | • The duration of high flows above Q95 will extend as long as 20 consecutive days over the winter<br><br>• Summer high flows will decrease | • Peak annual contribution from snowpack will shift from May to April<br><br>• Rain will increase its contribution to inflow during the winter | • Decrease in demand for energy over the winter<br><br>• Increase in demand for energy over the summer and fall |
| Opportunity - adaptation | Groupe E will want to ensure that they can capture and turbinate the increase in inflow over winter, when energy prices are typically at their highest.<br><br>Summer mean flows cannot be relied upon for energy supply. Groupe E may want to diversify their energy mix and/or develop their trading department. | When negotiating with water lease authorities, Groupe E can show the projections of low flow under the influence of climate change. This may allow them to negotiate a reduced water fee as well as residual flow requirements that do not impede their operations. | By adopting a flexible operating strategy, Groupe E could capitalize on the increase of high flows over winter, when energy prices are typically at their highest.<br><br>Summer high flows cannot be relied upon to offset drought. Groupe E may want to diversify their energy mix and/or develop their trading department to ensure that they meet their consumer's needs. | As spring runoff derived from snowpack becomes increasingly less reliable, Groupe E may want to increase their ability to capture high intensity rain events when they do happen (e.g. consecutive days above Q95 are shown to increase over winter). Developing the interaction between those in trading and meteorological forecasting may help to ensure the high-priced sale of electricity in advance of a high intensity event. | Groupe E will need to supply their consumer's increasing energy demand during the summer and fall with simultaneous reduced inflow entering their reservoirs. To offset this loss, they may want to adopt flexible operating strategies to capitalize on high electricity prices as well as to diversify their energy mix. |





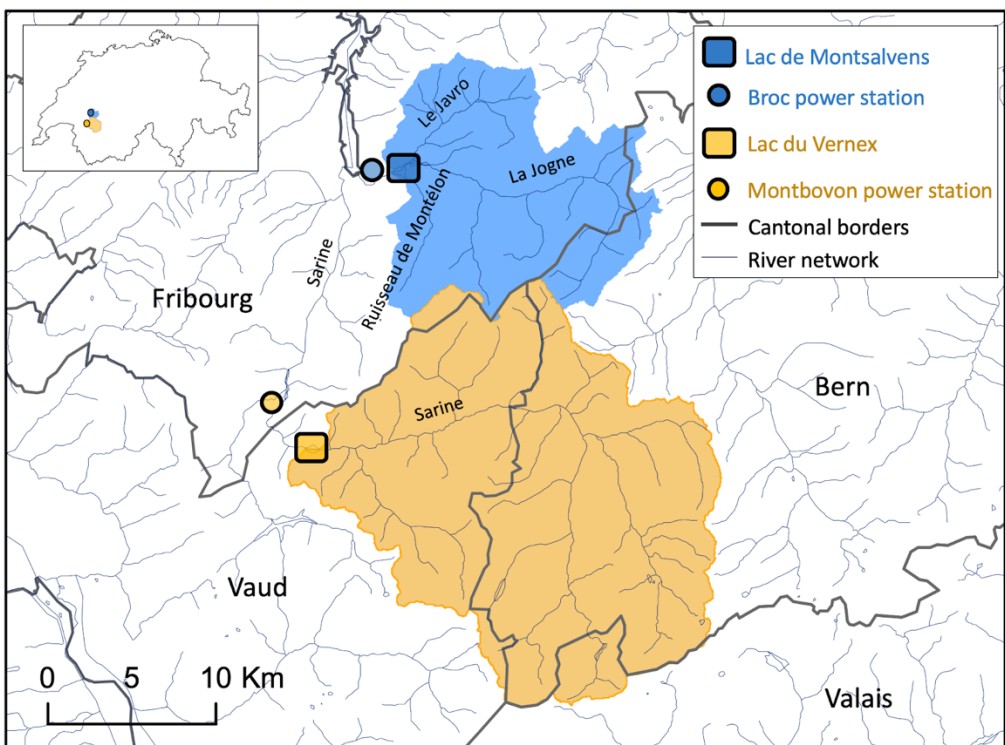

**Figure 1: Map of the two study catchments Montsalvens (blue) and Vernex (orange). The river network is shown in blue (dataset
provided by the Swiss Federal Office for the Environment, FOEN), the cantons are labeled and dark gray lines depict the cantonal
boundaries. The major river tributaries to the reservoirs are also labeled. The inset shows the location of the catchments within
Switzerland.**



**Figure 2: Performances of calibration of HBV and bias correction treatment are shown for each index for the Vernex catchment. When observational data were not available, then only bias correction performance is shown (plots e and f). All simulated data cover the period of 01-01-1980 to 31-12-2009, except for Q_obs data which spans the period of 01-10-2008 to 31-08-2018. Plots a, b, g, h depict long-term monthly means.**

730





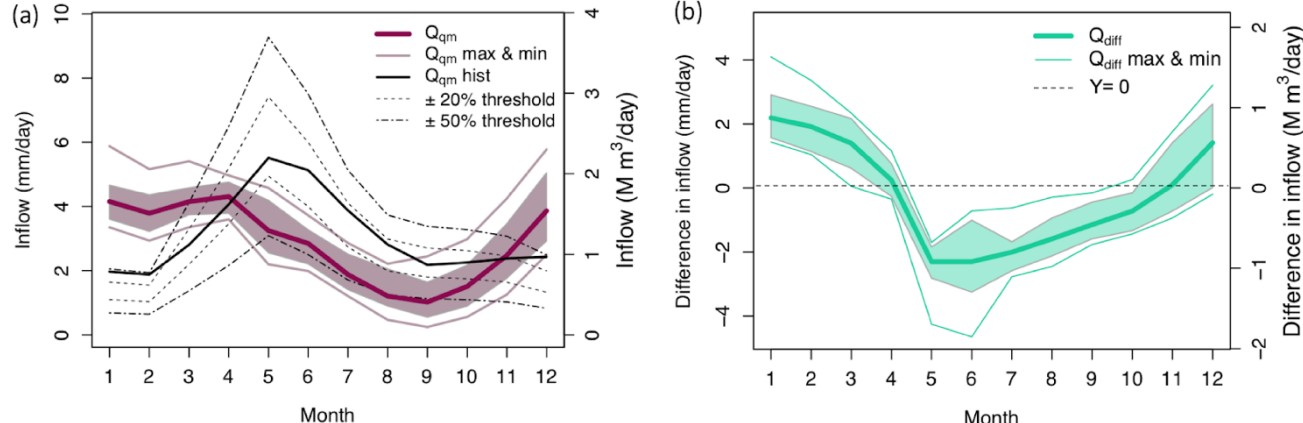

**Figure 3: (a) Long-term mean monthly inflow entering the Vernex reservoir for 1980-2009 ($Q_{qm}$ hist) and for 2070-2099 ($Q_{qm}$, RCP 8.5). The mean (solid lines) and likely range (shaded areas) are shown, where the likely range represents two thirds of all 660 simulations. The two thresholds are based on the mean of the simulations forced by observed climate data ($Q_{ref}$ over the period of 1980-2009). (b) Long-term mean monthly change in inflow (2070-2099 with respect to 1980-2009) for the Vernex catchment.**





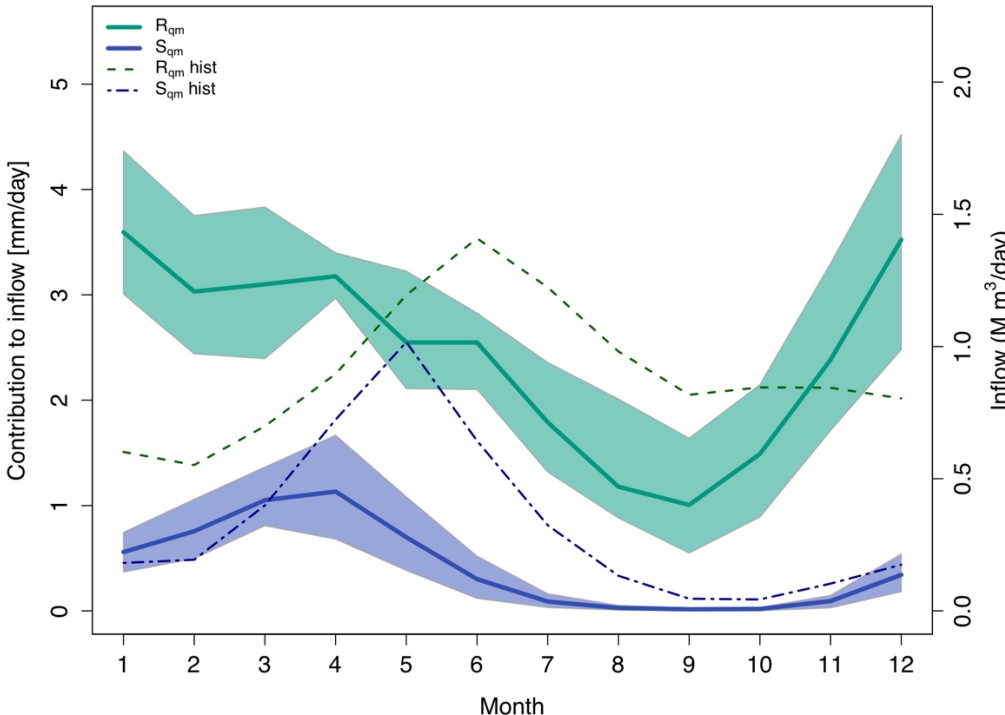

740

**Figure 4: Mean monthly contribution of rain (R, green) versus snow (S, blue) to inflow entering the Vernex reservoir. Two periods are compared: 1980-2009 ($R_{qm}$ and $S_{qm}$ hist) and 2070-2099 ($R_{qm}$ & $S_{qm}$). All projections shown are simulations under RCP 8.5. The mean (solid lines) and likely range (shaded areas) are shown, where the likely range represents two thirds of all 660 simulations. The dashed lines indicate the mean of the reference simulations.**

745





**Figure 5: (a) Boxplots showing low flow (Q5) and (b) high flow (Q95) indices, where the historical period (gray boxplots; 1980-2009) is compared to the future period (purple boxes; 2070-2099) for inflows entering the Vernex reservoir. All projections shown are for RCP 8.5. For each index, an associated ± 50% threshold is designated by a shaded area. These thresholds are based on the mean of simulations when forced by observed climate data (Q_ref) over the period of 1980-2009.**



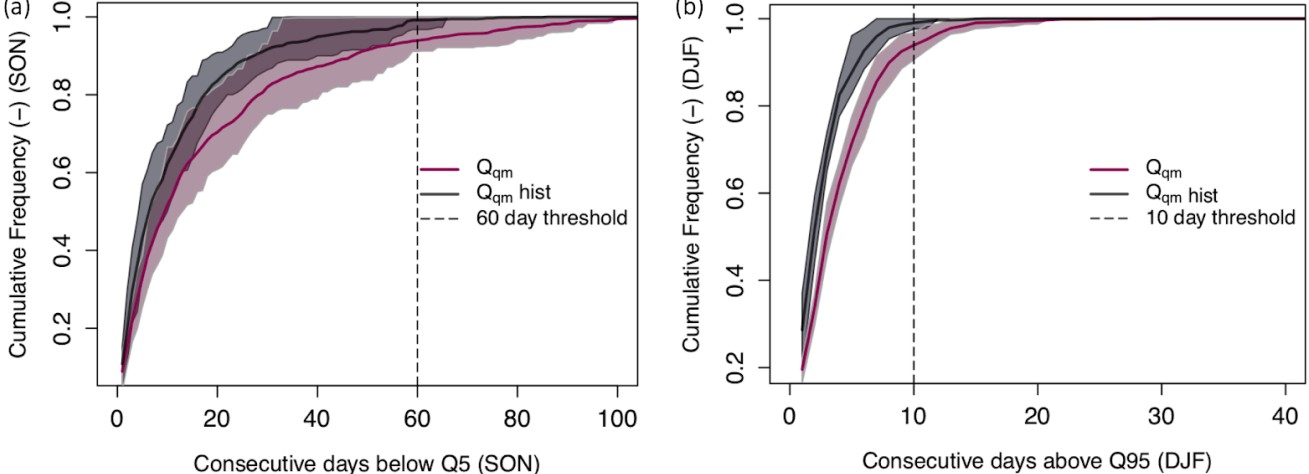

**Figure 6: Cumulative distribution functions (CDFs) are shown, where the historical period (1980-2009; gray) is compared to the future period (2070-2099; purple) for the Vernex catchment. (a) CDFs for consecutive days below Q5 are shown for the SON season, and a 60-day threshold is indicated by a black dashed line. (b) CDFs of the consecutive days above Q95 are shown for the season of DJF, and a 10-day threshold is shown by a black dashed line. Instances where the simulations exceed their associated threshold represent a level of change that is of interest to Groupe E. The mean (solid lines) and likely range (shaded areas) are shown, where the likely range represents two thirds of all 660 simulations.**



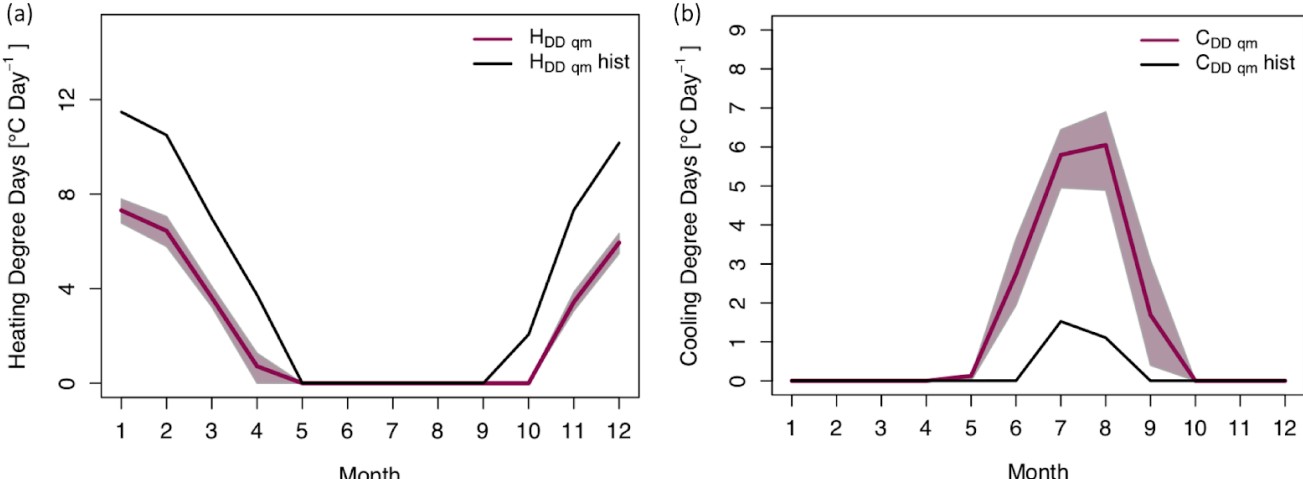

**Figure 7: Mean monthly (a) H$_{DD}$ and (b) C$_{DD}$ for the Canton of Geneva. The mean of the historical simulations (1980-2009; gray) are compared to the future simulations under the influence of RCP 8.5 climate change scenario (2070-2099; purple). The mean** 765 **(solid lines) and likely range (shaded areas) are shown, where the likely range represents two thirds of all 660 simulations. Group E prescribed thresholds of 13 °C and 18.3°C to compute H$_{DD}$ and C$_{DD}$, respectively.**

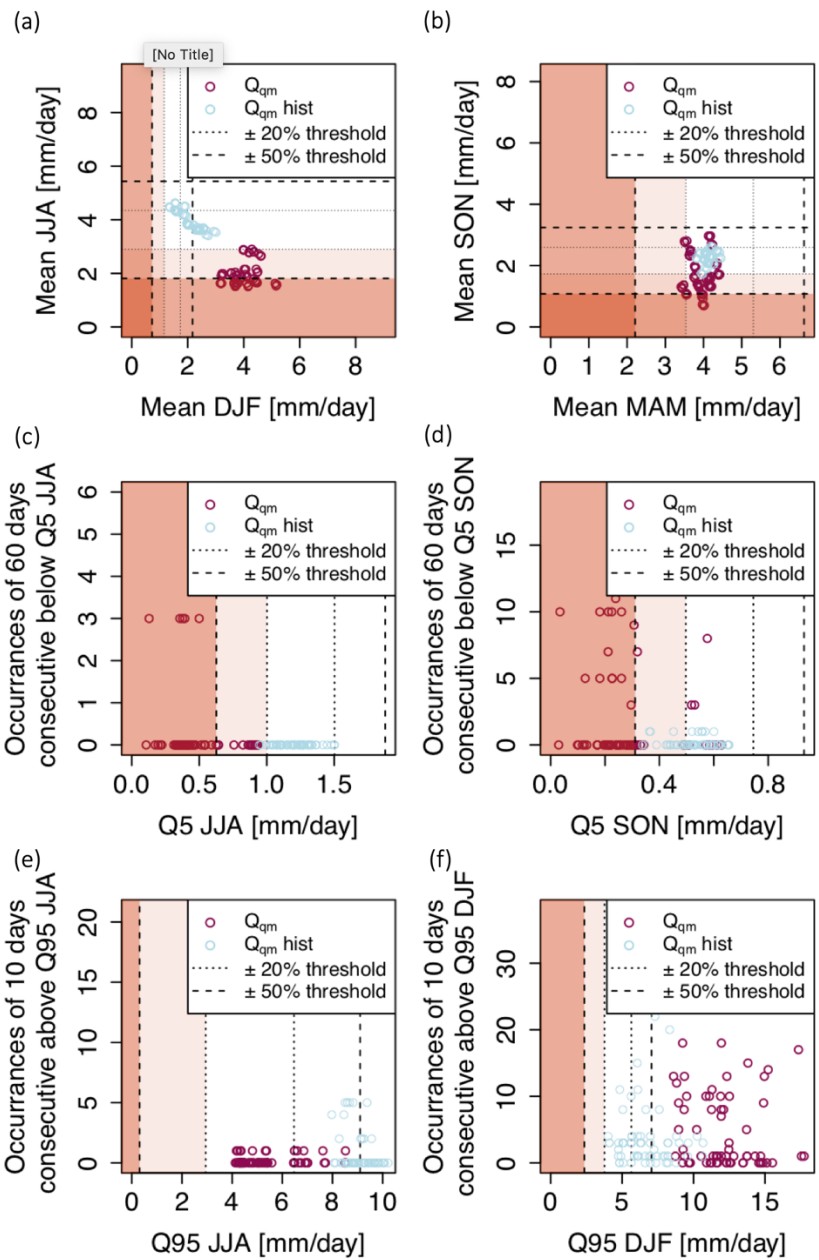

**Figure 8: Situations leading to the greatest stress on Groupe E's operations are depicted by the comparisons of low flow, high flow, and seasonal flow indices for the Vernex catchment. Two time periods are compared as left and right columns: (a-c) 1980-2009 and (d-f) 2070-2099. Plots (d-f) show simulations under the influence of RCP8.5. Plots (a) and (b) depict seasonal flows: mean winter flow (DJF) versus mean summer flow (JJA). Plots (c) and (d) depict low flows: Q5 summer flows (JJA) versus occurrences of consecutive days below Q5. Plots (e) and (f) depict high flows: Q95 winter flows (DJF) versus occurrences of consecutive days above Q95. For all plots, two thresholds are included: ± 20 and ±50 %, which were provided by Groupe E. Shading from white to progressively darker red tones indicates the least (white) to greatest (dark red) levels of stress placed on Groupe E's operations based on the relationship between the indices.**