# Peer review of "Risks and opportunities for a Swiss hydroelectricity company in a changing climate"

_Hydrology and Earth System Sciences, 2019_

## Referee Comment (RC1) · Anonymous Referee #1 · 18 Nov 2019

General comments

This study presents a case study focused on estimation of future changes in the streamflow and energy demand for two catchments in Switzerland. The main objective is to explore and evaluate the future inflows to two reservoirs used for hydropower production in order to support adaptation to climate change, and in particular, the negotiation of a new water concession.

The manuscript is clearly written and has a good structure, however I missed some more specific formulation of the scientific context and contribution. The link between science and engineering practice is interesting, but I wonder to what extent is the presented material in line with the scope of HESS. In its current form, the manuscript reads as a technical report and case study and does not provide clear formulations

and demonstrations of the novel scientific contribution. If this is not a key requirement for publication in HESS (I'm a bit confused by reading the current scope of HESS) then I recommend to publish it, if not a substantial revision improving the formulation/demonstration of novel scientific contribution is needed. It will be very interesting to see some more quantitative assessment of the steps used in the workflow (e.g. definition, evaluation of vulnerabilities) and how do those reduce the uncertainty (e.g. how did the multiple rounds of feedbacks quantitatively impact the projections and reduce the associated risks).

---

## Referee Comment (RC2) · Franco Romerio (Referee) · 8 Dec 2019

The article "Risks and opportunities for a Swiss hydropower company in a changing climate" addresses relevant questions of scientific and practical nature. It can be published after a careful revision. The main added value is given by the definition of a set of hydrological indices related to the Group E's vulnerability (well summarized in table 2). Of particular significance are the seasonal changes in inflow distribution. On this basis, the risks and opportunities for the operator are estimated. However, there are some drawbacks that should be considered by the authors, that I summarize below. The analysis focuses on water inflows and energy demand, but doesn't take into consideration other important characteristics of the energy turnaround and the opening of the power market to competition. Furthermore, as recognized by the authors, the analysis of the electricity demand is limited to the climate drivers. For instance, the authors point out "Under climate change, as flexibility decreases and energy demand likely increases due to heat waves [. . .], Groupe E stated that they would consider acquiring new sources of energy production to compensate for this loss." However, Groupe E forgets that very likely its customers will install solar panels and perhaps batteries for the storage of electricity at home. Its market will therefore change quite deeply. I understand that the authors can't develop this issue, which has not been studied. But they can't ignore it; on the contrary, they should emphasize it. Several times, the authors take as granted higher energy (in fact, electricity) prices in winter, "given that the winter period usually corresponds to higher energy prices". No doubt that this is a quite realistic scenario. But the problem is very complex and the future evolution of the market in Europe and Switzerland presents many uncertainties, as shown by a quite large literature. Some scenarios are not so favorable to hydropower. In any case, it is risky to assume a continuity between past and future. The authors state: "The figures we provide will help Groupe E determine the value of water in the future and the price they are willing to pay for the renewal of their concessions." I agree, but on two conditions: it must be recognized that an economic study of the value of water has not been carried out; one must be aware that only two drivers have been taken into consideration: water flows and energy demand. The collaboration with group E is the basis of this article. It produced interesting results. Group E must be thanked for its transparency. But my feeling is that the authors rely too much on the company's expertise. It looks like that ultimately it belongs to the company to decide if the authors' analyses are relevant or not. The authors should be a little more critic. Moreover, the company's judgments depend on its strategy, which is not presented in the article. The authors state: "This study illustrates the benefits of involving stakeholders in hydropower climate change impact studies". In fact, only one stakeholder was involved in this study, i.e. a power company. It would have been more interesting to highlight the perception of different stakeholders (public bodies, environmental organizations, local communities, etc.) on these issues, as well as their convergences and divergences. The authors stress the

importance of their results in the context of the negotiations of the concessions renewal, residual flows and water fees. However, in order to develop a strategy, one has to understand the point of view of the main stakeholders involved. For instance, I imagine that the Groupe E's request that "residual flow requirements should not increase or find a middle ground given the future behavior of low flows entering their reservoirs", will be challenged by stakeholders primarily concerned by environmental issues.

---

## Author Comment (AC1) · 17 Jan 2020

**Authors' response to the interactive comment of referee #1 on hessd-2019-475**
Risks and opportunities for a Swiss hydropower company in a changing climate

We thank the anonymous reviewer for his/her general positive evaluation of our manuscript, and the helpful comments regarding the scope of the paper. Below we respond to the reviewer (*in blue*). We appreciate the comments by the reviewer, which will be used to help clarify the objectives of our manuscript.

Referee #1:

This study presents a case study focused on estimation of future changes in the streamflow and energy demand for two catchments in Switzerland. The main objective is to explore and evaluate the future inflows to two reservoirs used for hydropower production in order to support adaptation to climate change, and in particular, the negotiation of a new water concession.

The manuscript is clearly written and has a good structure, however I missed some more specific formulation of the scientific context and contribution. The link between science and engineering practice is interesting, but I wonder to what extent is the presented material in line with the scope of HESS. In its current form, the manuscript reads as a technical report and case study and does not provide clear formulations and demonstrations of the novel scientific contribution. If this is not a key requirement for publication in HESS (I'm a bit confused by reading the current scope of HESS) then I recommend to publish it, if not a substantial revision improving the formulation/ demonstration of novel scientific contribution is needed.

We submitted our manuscript to HESS because the journal aims to serve not only the hydrological science community but also water managers. Our article was submitted with the intention of addressing the third scope of HESS outlined on the journal website (https://www.hydrology-and-earth-system-sciences.net/about/aims_and_scope.html), namely "the study of interactions with human activity of all the processes, budgets, fluxes, and pathways, and the options for influencing them in a sustainable manner, particularly in relation to floods, droughts, desertification, land degradation, eutrophication, and other aspects of global change."

As stated in the introduction and throughout the manuscript, most climate change impact studies adopt a top-down approach (i.e., the end-users are hardly explicitly considered nor are the end-users involved in the modeling framework). By collaborating with a hydropower company, we were able to identify the key climate-change-related challenges of their operations. We believe that our manuscript stands out because of the bottom-up approach employed and the resulting selection of indices, which were chosen to address Groupe E's vulnerabilities (summarized in Table 2). By utilizing a bottom-up approach, our study highlights that the hydrological opportunities and risks associated with reservoir management in a changing climate depend on a range of factors beyond those covered by traditional impact studies. Given that many hydropower companies will soon enter the renegotiation process of their concessions, this work is timely, as it illustrates that these negotiations could benefit from a bottom-up climate change impact assessment. However, we agree with the referee that the novel contributions of our study could be stated more clearly. We will hence revise the text, including the abstract.

It will be very interesting to see some more quantitative assessment of the steps used in the workflow (e.g. definition, evaluation of vulnerabilities) and how do those reduce the uncertainty (e.g. how did the multiple rounds of feedbacks quantitatively impact the projections and reduce the associated risks).

We found this interesting as well and will expand upon the ways in which the feedback received by Groupe E helped to focus the design of our study. We will discuss in Section 5.2 how the multiple rounds of feedback helped to direct our attention and efforts to certain aspects of the modelling chain (e.g. calibration of HBV, selection of climate models) and how they influenced the analysis and the visualisation of the simulations (e.g. selection of streamflow indices, inclusion of thresholds and uncertainty estimates in the figures).

---

## Author Comment (AC2) · 17 Jan 2020

**Authors' response to the interactive comment of referee #2 on hessd-2019-475**
Risks and opportunities for a Swiss hydropower company in a changing climate

We thank Prof. Dr. Franco Romerio for his helpful comments regarding the uncertainties of the European and Swiss energy markets and for lending his insight to improve our manuscript. Below we respond to his comments (*in blue*). We appreciate the time taken for this review, and we believe that our incorporation of his comments will improve the uncertainty discussion within the manuscript.

Referee #2:

The article "Risks and opportunities for a Swiss hydropower company in a changing climate" addresses relevant questions of scientific and practical nature. It can be published after a careful revision. The main added value is given by the definition of a set of hydrological indices related to the Group E's vulnerability (well summarized in table 2). Of significance are the seasonal changes in inflow distribution. On this basis, the risks and opportunities for the operator are estimated. However, there are some drawbacks that should be considered by the authors, that I summarize below.

The analysis focuses on water inflows and energy demand, but doesn't take into consideration other important characteristics of the energy turnaround and the opening of the power market to competition. Furthermore, as recognized by the authors, the analysis of the electricity demand is limited to the climate drivers. For instance, the authors point out "Under climate change, as flexibility decreases and energy demand likely increases due to heat waves […], Groupe E stated that they would consider acquiring new sources of energy production to compensate for this loss." However, Groupe E forgets that very likely its customers will install solar panels and perhaps batteries for the storage of electricity at home. Its market will therefore change quite deeply. I understand that the authors can't develop this issue, which has not been studied. But they can't ignore it; on the contrary, they should emphasize it. Several times, the authors take as granted higher energy (in fact, electricity) prices in winter, "given that the winter period usually corresponds to higher energy prices". No doubt that this is a quite realistic scenario. But the problem is very complex and the future evolution of the market in Europe and Switzerland presents many uncertainties, as shown by a quite large literature. Some scenarios are not so favorable to hydropower. In any case, it is risky to assume a continuity between past and future.

We agree with your statements concerning the limitations of our study, namely that an economic study of the future value of water was not carried out. We recognize that there is room for expansion of this work, and indeed this is something that we are hoping will inspire other researchers and end-users. Within the next version of this article, we will further discuss the limitations of our study, in particular that we do not explicitly consider the economic uncertainties related to the European and Swiss energy markets. Groupe E itself has been clear about these uncertainties and about the fact that our projections are only a piece of the puzzle. We will make this clearer in the revised version of the manuscript. Importantly, Groupe E also asked for more projections with expanded indices and for additional reservoirs, which demonstrates that these projections are useful for their concession negotiation process, despite their uncertainties and limitations. We note that the need for projections during the concession negotiation process is also highlighted by Tonka (2015)*, who stated that there is a "striking lack of attention paid to climate change impacts on water resources availability in relicensure procedures in the USA. In Switzerland, studies on this topic have been undertaken only recently and climate change projections have not yet affected the content of renewed concessions." We will stress this too in the revised version.

*Tonka, L., Hydropower license renewal and environmental protection policies: a comparison between Switzerland and the USA. Regional Environmental Change 15, 539-548 (2015) doi:10.1007/s10113-014-0598-8

The authors state: "The figures we provide will help Groupe E determine the value of water in the future and the price they are willing to pay for the renewal of their concessions." I agree, but on two conditions: it must be recognized that an economic study of the value of water has not been carried out; one must be aware that only two drivers have been taken into consideration: water flows and energy demand.

We agree entirely, and perhaps our statements regarding this were overlooked. Within Section 2.1.1 we state, "During concession negotiations, Groupe E representatives stated that the following would be considered (i) the development of the energy market and competitors, (ii) the projected supply of water resources, (iii) changes in energy demand, and (iv) costs associated with adhering to new environmental standards. This study focuses on the estimation of future water resources (point ii) and providing preliminary insights into future energy demand (point iii)."

Within the next version of this article, we will make this point more explicit (likely by moving the text to the introduction). We agree that the incorporation of an economic study of the value of water into this project would provide additional insights and therefore improved guidance to water managers. We will expand Section 5.4, which is currently dedicated to the discussion of this point.

The collaboration with group E is the basis of this article. It produced interesting results. Group E must be thanked for its transparency. But my feeling is that the authors rely too much on the company's expertise. It looks like that ultimately it belongs to the company to decide if the authors' analyses are relevant or not. The authors should be a little more critical.

We agree and will change the text to be more critical of how our projections may be used by Group E.

Moreover, the company's judgments depend on its strategy, which is not presented in the article.

Although Groupe E has been very transparent with us, their future strategies are not meant to be made public. Specific language about their future investments was purposely removed from the manuscript. We will work with our two Groupe E contacts (who are co-authors on this manuscript) in order to expand our discussion of Groupe E's strategies where we can. We will also make clearer that Group E's future strategy is confidential and cannot be fully disclosed in this paper.

The authors state: "This study illustrates the benefits of involving stakeholders in hydropower climate change impact studies". In fact, only one stakeholder was involved in this study, i.e. a power company. It would have been more interesting to highlight the perception of different stakeholders (public bodies, environmental organizations, local communities, etc.) on these issues, as well as their convergences and divergences.

We agree that the inclusion of more stakeholders, public bodies, and environmental organizations would be an exciting and valuable way to move forward. It is not within the scope of this current project to include these additional complexities to the study. Nonetheless, this point will be elaborated upon within Section 5.4. This is indeed an important next step, and it so happens that the first author has a new project already underway to explore this level of collaboration and complexity within streamflow and water demand forecasting (within Australia).

The authors stress the importance of their results in the context of the negotiations of the concessions renewal, residual flows and water fees. However, in order to develop a strategy, one has to understand the point of view of the main stakeholders involved. For instance, I imagine that the Groupe E's request that "residual flow requirements should not increase or find a middle ground given the future behavior of low flows entering their reservoirs", will be challenged by stakeholders primarily concerned by environmental issues.

Our aim was to give Groupe E a first insight into the impacts of climate change on the inflow entering their reservoir and into possible changes in energy demand based on air temperature. It is true that Groupe E will likely face challenges from stakeholders during the negotiation process. We will develop our text further to reflect the need for future research that incorporates the viewpoints and concerns of all main stakeholders. We will also reiterate that the challenge of concession negotiations has multiple facets and that our focus in this study is to support these negotiations with climate change impact assessments using a bottom-up methodology.

---

## Author Response (AR1)

Journal: Hydrology Earth System Sciences

Title: Risks and opportunities for a Swiss hydroelectricity company in a changing climate

Authors: Kirsti Hakala, Nans Addor, Thibault Gobbe, Johann Ruffieux, Jan Seibert

Dear Prof. Dr. Markus Hrachowitz, Prof. Dr. Franco Romerio, and Anonymous Reviewer,

We thank the associate editor and our two reviewers for their insightful comments and for lending their perspective to our study. To address the comments shared by Prof. Dr. Hrachowitz and the anonymous reviewer, we reframed our scientific questions so that they are now more testable and of more general relevance. We bolstered our discussion of the limitations of our study to address the requests of the second reviewer, Prof. Dr. Romerio. We also reduced the number of times we mentioned of Groupe E so that our main messages can be more broadly received by our readers. Our detailed response to the reviewers' and editor's comments are included on the following pages, with our responses shown in blue. We believe that the incorporated feedback has helped us fine-tune and improve our manuscript. We hereby resubmit the revised version of our manuscript for publication in Hydrology Earth System Sciences.

On behalf of all co-authors-

Sincerely,

Kirsti Hakala

Comments from the Editor (Prof. Dr. Markus Hrachowitz)
* * *
As you have seen the two reviewers highly appreciate your contribution. I agree with this assessment and I think that your manuscript does have the potential to become an important reference. However, the reviewers also raise a few relevant concerns and provide several excellent suggestions to strengthen the manuscript. From my perspective the most relevant point is the framing of the manuscript as stressed by reviewer #1. While the analysis presented does indeed fall within the scope of HESS, the study comes too much across as a technical report. I understand that it is a thin line between what you are trying to do and technical reports. Yet, I feel the manuscript needs to be a bit more balanced towards science aspects. This could be relatively easily done by sharpening the science question, the research hypotheses to be tested and the discussion of the scientific context.

We agree with this comment and have subsequently reformulated our research questions on lines 76-85 (line numbers correspond to when track changes are turned off). The research questions now read:

1. Climate change impacts on water resources are already broadly described by the scientific literature and reports published by public entities (e.g., environmental agencies). While this broad-scale information is available to hydroelectricity companies, is it adequate to support their negotiations for concession renewal?
2. Future climate change impacts are uncertain and are typically communicated using an ensemble of simulations. How well do stakeholders incorporate this uncertainty into their decision-making process on adaption strategies?
3. Future reservoir profitability depends on a wide range of economic and environmental factors. How can projections focused on the availability of water resources be leveraged in the negotiation process of a reservoir concession, and what are their limitations?

Our discussion section addresses each of our research questions: RQ 1 = Section 5.1 (& 5.4 discusses limitations), RQ 2 = Section 5.2, and RQ 3 = Section 5.3.

It may also be very valuable to discuss the limitations inherent in the assumptions taken and their (potential) effects on the results/interpretation in more detail as requested by reviewer #2.

We appreciate the insight that Prof. Dr. Franco Remerio offered. We expanded our discussion of our limitations within Section 5.4 (please see lines 593 – 604, which now reads, "Concession negotiations have many facets and although hydrological changes are important, they only partially determine the profitability of hydropower operations. This study focused on hydroclimatic changes using a range of streamflow indices. We did not account for the uncertainties related to the development of the European or Swiss electricity market. Instead we used a simple method to estimate future electricity demand solely based on air temperature. This study nevertheless points out that despite the uncertainties involved, quantifying the supply of future water resources and providing an estimate of changes to demand (based on changes to air temperature) improves the information currently available to electricity managers, and is useful for their concession negotiations."

In order to make our contributions clear, lines 83-87 state, "The representatives stated that they expect the following to be considered during concession negotiations (i) the development of the electricity market and competitors, (ii) the projected supply of water resources, (iii) changes in electricity demand, and (iv) costs associated with adhering to new environmental standards. This study focuses on the

estimation of future water resources (point ii) and provides preliminary insight into future electricity demand (point iii)." This was already mentioned in the previous version of this manuscript, but we moved this text to a more prominent location to avoid confusion.

We also reigned in our text where we make assumptions about future economic principles and user behavior and point out that we are assuming a continuity between past and present. For example, lines 415-417 read, "Therefore, these changes could allow Groupe E to capitalize on generally higher electricity prices in winter (assuming that electricity prices remain higher in winter than in summer) resulting in a potential increase in profits for this season."

Now while I also understand that Groupe E thankfully made their data and knowledge available to allow a bottom-up approach here, Groupe E will also benefit from this study. In that sense, I believe the repeated explicit mentioning of Groupe E does not do service to this manuscript. The work would benefit from less emphasis on the name of the industry partner and a more generic presentation of the work.

We agree and have reduced the number of times that we specifically mention Groupe E throughout the manuscript. Direct references to Groupe E still occur where necessary, but overall we focused on broadening our main points so that they can be received by a wider audience.

Comments from Reviewer #1 (Anonymous)
* * *
This study presents a case study focused on estimation of future changes in the streamflow and energy demand for two catchments in Switzerland. The main objective is to explore and evaluate the future inflows to two reservoirs used for hydropower production in order to support adaptation to climate change, and in particular, the negotiation of a new water concession.

The manuscript is clearly written and has a good structure, however I missed some more specific formulation of the scientific context and contribution. The link between science and engineering practice is interesting, but I wonder to what extent is the presented material in line with the scope of HESS. In its current form, the manuscript reads as a technical report and case study and does not provide clear formulations and demonstrations of the novel scientific contribution. If this is not a key requirement for publication in HESS (I'm a bit confused by reading the current scope of HESS) then I recommend to publish it, if not a substantial revision improving the formulation/ demonstration of novel scientific contribution is needed.

Thank you again for this comment – this feedback inspired us to restructure our research questions so that they are more testable. In addition, we incorporated this concept into our discussion section, which we believe helped focus our text on more scientific aspects. This point was also reiterated by the editor – please see our response to Prof. Dr. Markus Hrachowitz for a more elaborate response.

It will be very interesting to see some more quantitative assessment of the steps used in the workflow (e.g. definition, evaluation of vulnerabilities) and how do those reduce the uncertainty (e.g. how did the multiple rounds of feedbacks quantitatively impact the projections and reduce the associated risks).

Currently, Section 2.3 explains how our interactions with stakeholders resulted in a selection of indices and thresholds that allowed us to evaluate their vulnerabilities under climate change. The first paragraph of Section 5.3 describes how a stakeholder-centered approach influenced how we visualized uncertainty in our figures. We included additional text on lines 237-241 to explain how this approach influenced our selection of climate models, given the focus of this project on hydrological changes relevant for hydropower operations.

Comments from Reviewer #2 (Prof. Dr. Franco Romerio)
* * *
The article "Risks and opportunities for a Swiss hydropower company in a changing climate" addresses relevant questions of scientific and practical nature. It can be published after a careful revision. The main added value is given by the definition of a set of hydrological indices related to the Group E's vulnerability (well summarized in table 2). Of significance are the seasonal changes in inflow distribution. On this basis, the risks and opportunities for the operator are estimated. However, there are some drawbacks that should be considered by the authors, that I summarize below.

The analysis focuses on water inflows and energy demand, but doesn't take into consideration other important characteristics of the energy turnaround and the opening of the power market to competition. Furthermore, as recognized by the authors, the analysis of the electricity demand is limited to the climate drivers. For instance, the authors point out "Under climate change, as flexibility decreases and energy demand likely increases due to heat waves […], Groupe E stated that they would consider acquiring new sources of energy production to compensate for this loss." However, Groupe E forgets that very likely its customers will install solar panels and perhaps batteries for the storage of electricity at home. Its market will therefore change quite deeply. I understand that the authors can't develop this issue, which has not been studied. But they can't ignore it; on the contrary, they should emphasize it. Several times, the authors take as granted higher energy (in fact, electricity) prices in winter, "given that the winter period usually corresponds to higher energy prices". No doubt that this is a quite realistic scenario. But the problem is very complex and the future evolution of the market in Europe and Switzerland presents many uncertainties, as shown by a quite large literature. Some scenarios are not so favorable to hydropower. In any case, it is risky to assume a continuity between past and future.

We agree with your statements concerning the limitations of our study, namely that an economic study of the future value of water was not carried out. We recognize that there is room for expansion of this work, and indeed this is something that we are hoping will inspire other researchers and end-users. We have expanded Section 5.4, which points out the limitations of our study and the need to incorporate economic principles as a next step. We also reigned in our text where we make assumptions about future economic principles and user behavior and point out that we are assuming a continuity between past and present. For example, lines 415-417 read, "Therefore, these changes could allow Groupe E to capitalize on generally higher electricity prices in winter (assuming that electricity prices remain higher in winter than in summer) resulting in a potential increase in profits for this season."

Also, please see our reformulated conclusions section, which has been tempered by this point.

Despite the limitation of our study, we still believe that this study is timely and useful. Our conclusions section states the following on lines 656-562, "This study is timely as many electricity managers are

currently faced with renegotiating their water concessions in the context of climate change and an uncertain electricity market. Yet, as Tonka (2015) notes, there has been a 'striking lack of attention paid to climate change impacts on water resources availability in relicensure procedures'. We show that although many uncertainties exist, given the multi-decade length of a concession, it is crucial for climate change to be considered at the onset of concession negotiations."

*Tonka, L., Hydropower license renewal and environmental protection policies: a comparison between Switzerland and the USA. Regional Environmental Change 15, 539-548 (2015) doi:10.1007/s10113-014-0598-8*

The authors state: "The figures we provide will help Groupe E determine the value of water in the future and the price they are willing to pay for the renewal of their concessions." I agree, but on two conditions: it must be recognized that an economic study of the value of water has not been carried out; one must be aware that only two drivers have been taken into consideration: water flows and energy demand.

We agree entirely, and we have moved our statements regarding this to a more prominent location – the introduction. Lines 83-87 read, "The representatives stated that they expect the following to be considered during concession negotiations (i) the development of the energy market and competitors, (ii) the projected supply of water resources, (iii) changes in energy demand, and (iv) costs associated with adhering to new environmental standards. This study focuses on the estimation of future water resources (point ii) and providing preliminary insights into future energy demand (point iii)."

The collaboration with group E is the basis of this article. It produced interesting results. Group E must be thanked for its transparency. But my feeling is that the authors rely too much on the company's expertise. It looks like that ultimately it belongs to the company to decide if the authors' analyses are relevant or not. The authors should be a little more critical.

We agree and therefore expanded our discussions based on existing literature rather than solely relying on Groupe E to validate the usefulness of this study. Regarding our discussion of the future of electricity demand, please see the completely reformatted Section 5.1.4. Our discussion of changes to water volume and how this relates to hypothetical water fee systems is located in the last paragraph of Section 5.1.1 on lines 430-444, and is tempered by your 2018 paper* with Ludovic Gaudard.

*Gaudard, L., Voegeli, G. and Romerio, F.: Hydropower investment profitability under different water fee systems., 2018a.

Moreover, the company's judgments depend on its strategy, which is not presented in the article.

Although Groupe E has been very transparent with us, their future strategies are not meant to be made public. Specific language about their future investments was purposely removed from the manuscript. This point is addressed on lines 141-142 which now states, "Groupe E was very transparent throughout this collaboration, however Groupe E's future strategies are confidential and cannot be fully disclosed in this paper.

The authors state: "This study illustrates the benefits of involving stakeholders in hydropower climate change impact studies". In fact, only one stakeholder was involved in this study, i.e. a power company. It would have been more interesting to highlight the perception of different stakeholders (public bodies,

environmental organizations, local communities, etc.) on these issues, as well as their convergences and divergences.

We agree that the inclusion of more stakeholders, public bodies, and environmental organizations would be an exciting and valuable way to move forward. It is not within the scope of this current project to include these additional complexities to the study. Lines 613-617 now state, "New research projects would benefit from involving a wider range of stakeholders. A collaboration between hydrologists, economists, and stakeholders such as cantonal authorities, environmental interest groups, hydropower operations specialists, electricity market traders would help to support concession negotiations and to foster the sustainable development of hydropower."

The authors stress the importance of their results in the context of the negotiations of the concessions renewal, residual flows and water fees. However, in order to develop a strategy, one has to understand the point of view of the main stakeholders involved. For instance, I imagine that the Groupe E's request that "residual flow requirements should not increase or find a middle ground given the future behavior of low flows entering their reservoirs", will be challenged by stakeholders primarily concerned by environmental issues.

Our aim was to give Groupe E a first insight into the impacts of climate change on the inflow entering their reservoir and into possible changes in energy demand based on air temperature. It is true that Groupe E will likely face challenges from stakeholders during the renegotiation process. We therefore developed our text further to reflect the need for future research that incorporates the viewpoints and concerns of all main stakeholders (please our response to your previous point above). We also tempered our language where possible – for instance our discussion of low flows in Section 5.1.2 has been updated. Lines 458-462 now read, "
[revised manuscript text omitted]